



# Systematic Comparison of Vectorial Spherical Radiative Transfer Models in Limb Scattering Geometry

Daniel Zawada[1], Ghislain Franssens[2], Robert Loughman[3], Antti Mikkonen[4], Alexei Rozanov[5], Claudia Emde[6], Adam Bourassa[1], Seth Dueck[1], Hannakaisa Lindqvist[4], Didier Ramon[7], Vladimir Rozanov[5], Emmanuel Dekemper[2], Erkki Kyrölä[4], John P. Burrows[5], Didier Fussen[2], and Doug Degenstein[1]

[1]Institute of Space and Atmospheric Studies, University of Saskatchewan, Saskatchewan, Canada
[2]Royal Belgian Institute for Space Aeronomy, Brussels, Belgium
[3]Department of Atmospheric and Planetary Sciences, Hampton University, Hampton, Virginia, USA
[4]Finnish Meteorological Institute, Helsinki, Finland
[5]Institute of Environmental Physics, University of Bremen, Bremen, Germany
[6]Meteorological Institute, Ludwig-Maximilians-University, Munich, Germany
[7]HYGEOS, Lille, France

**Correspondence:** Daniel Zawada (daniel.zawada@usask.ca)

**Abstract.** A comprehensive inter-comparison of seven radiative transfer models in the limb scattering geometry has been performed. Every model is capable of accounting for polarisation within a fully spherical atmosphere. Three models (GSLS, SASKTRAN-HR, and SCIATRAN) are deterministic, and four models (MYSTIC, SASKTRAN-MC, Siro, and SMART-G) are statistical using the Monte Carlo technique. A wide variety of test cases encompassing different atmospheric conditions, solar geometries, wavelengths, tangent altitudes, and Lambertian surface reflectances have been defined and executed for every model. For the majority of conditions it was found that the models agree to better than $0.2\%$ in the single scatter test cases and better than $1\%$ in the multiple scatter scalar and vector test cases, with some larger differences noted at high values of surface reflectance in multiple scatter. For the first time in limb geometry, the effect of atmospheric refraction was compared among four models that support it (GSLS, SASKTRAN-HR, SCIATRAN, and SMART-G). Differences among most models in multiple scatter with refraction enabled was less than $1\%$, with larger differences observed for some models. Overall the agreement among the models with and without refraction is better than has been previously reported in both scalar and vector modes.

## 1 Introduction

The limb scattering measurement technique involves viewing through the side, the limb, of the atmosphere while measuring scattered sunlight (see Fig. 1). Measurements are performed in the ultraviolet, visible, and near infrared spectral ranges where





scattering of solar irradiance is the dominant source of measured radiation. Scattering occurs through Rayleigh scattering from the background atmosphere, as well as potential contributions of scattering from larger particles such as stratospheric aerosols and clouds. The signal is also affected through absorption by atmospheric constituents, typically by molecules such as ozone

or nitrogen dioxide in the ultraviolet and visible. In the near and short wave infrared, absorption is dominated by water vapour, methane, carbon dioxide, and molecular oxygen.

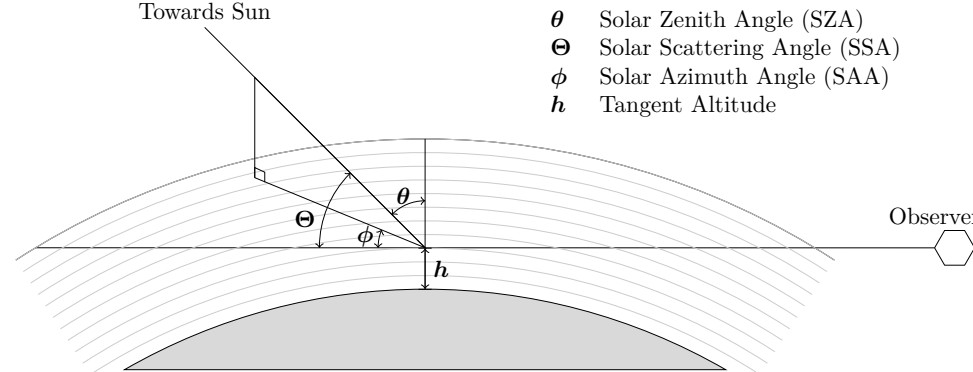

**Figure 1.** The limb viewing geometry and definitions of solar zenith angle, solar azimuth angle, solar scattering angle, and tangent altitude. The tangent altitude and solar zenith angles are defined relative to the un-refracted tangent point. Figure adapted from Zawada et al. (2015).

Several satellite-based limb scattering instruments have flown in the past few decades. Notably, the Optical Spectrograph and InfraRed Imager System (OSIRIS, Llewellyn et al., 2004) was launched on-board the Swedish satellite Odin (Murtagh et al., 2002) in 2001, the SCanning Imaging Absorption SpectroMeter for Atmospheric CHartographY (SCIAMACHY, Bovensmann, 1999) instrument on-board Envisat in 2002, and the Ozone Mapping and Profiler Suite Limb Profiler (OMPS-LP, Flynn et al.,

2006) on-board Suomi-NPP in 2011. Two versions of the primarily solar occultation instrument the Stratospheric Aerosol and Gase Experiment (SAGE), SAGE III-M (Mauldin et al., 1998) on Meteor-3M in 2001, and SAGE III-ISS (Cisewski et al., 2014) on the International Space Station (ISS) in 2016, have capability to make limb scatter measurements. The stellar occultation instrument, Global Ozone Monitoring by Occultation of Stars (GOMOS, Kyrölä et al., 2004), is also capable of taking limb

scatter measurements. OMPS-LP is planned to be re-launched on-board the JPSS-2 satellite in 2022, and a new instrument, the Atmospheric Limb Tracker for Investigation of the Upcoming Stratosphere (ALTIUS, Fussen et al., 2019) is currently under development by the European Space Agency for a planned 2024 launch.

Vertical profiles of limb scattering spectra can be inverted to obtain distributions of atmospheric constituents with spectral absorption or scattering features. These include but are not limited to: stratospheric aerosol (Bourassa et al., 2007; Von Savigny

et al., 2015), ozone (Roth et al., 2007; Degenstein et al., 2009; Rault and Spurr, 2010; Arosio et al., 2018), nitrogen dioxide (Butz et al., 2006; Sioris et al., 2017), water vapor (Rozanov et al., 2011b), and bromine oxide (McLinden and Bourassa, 2010; Rozanov et al., 2011a). Inversion of these spectra requires a Radiative Transfer Model (RTM) capable of simulating



the observed radiance, including all relevant physical effects for the constituent of interest. The accuracy of the RTM directly influences the accuracy of the retrieved profile.

All of the aforementioned retrieval methods use RTMs operating in scalar mode, where only the intensity, $I$, is computed rather than the full Stokes vector of the observed radiance. This is known to be a good approximation when the instrument itself is designed to be polarization insensitive. While there do exist errors in the intensity from neglecting polarization (Mishchenko et al., 1994), they tend to cancel in various normalization schemes used by the retrieval (Loughman et al., 2005). However, ALTIUS as well as similar instrument concepts (Elash et al., 2017; Kozun et al., 2020), are designed to measure a linear

polarized signal rather than the raw intensity. For these instruments it is required to use an RTM that is capable of simulating the full Stokes vector.

Several studies have been performed which inter-compare the accuracy of polarized RTMs (e.g. Emde et al., 2015, 2018), however these have been focused on viewing angles of at most 80° (a viewing angle of 90° would be limb viewing). For viewing geometries other than limb and occultation it is common to use the plane parallel assumption, which is generally not

applicable in the limb geometry. The approximative spherical approach, where single scattered radiance is calculated using a spherical atmosphere and the multiple scatter signal is approximated with a plane-parallel model, has been shown to have systematic errors in the limb viewing geometry (Loughman et al., 2004; McLinden and Bourassa, 2010). Recently, Korkin et al. (2020) have extended some of these results to a fully spherical atmosphere, but the scope of the project was limited to scalar radiative transfer.

The most comprehensive previous inter-comparison focusing on the limb scattering geometry was performed by Loughman et al. (2004). The deterministic models Gauss-Seidel Limb Scattering (GSLS), CDI and CDIPI (Rozanov et al., 2001,  which are solvers implemented in SCIATRAN), and the statistical models Siro (Oikarinen et al., 1999) and MCC++ (Postylyakov, 2004) were considered. It was found that the statistical models generally agree to within 1.5% in the total scattering case, while the deterministic spherical models, CDIPI and GSLS, agree with the statistical models at the 2–4% level. While comparison

of the full Stokes vector was included for models that supported it, it was not the primary goal of the study. The results of this study have been used in benchmarking newly developed RTMs, such as SASKTRAN (Bourassa et al., 2008), or to evaluate new updates or features of RTMs as was done for GSLS (Loughman et al., 2015).

This study serves to both update the state of inter-comparison of RTMs in limb scattering geometry and to improve on it in several ways. Firstly, all participating RTMs simulate polarization in the atmosphere and provide full Stokes vectors

which are compared, these results are of significant importance for the upcoming ALTIUS mission. Secondly, the treatment of stratospheric aerosols is updated to use Mie scattering solution rather than a Henyey-Greenstein phase function; the Mie scattering treatment is more representative of the current state of limb retrievals (e.g. Rieger et al., 2019; Malinina et al., 2019; Taha et al., 2020). In addition, simulations including atmospheric refraction are included for the first time.

Descriptions of each model is presented in Sec. 2, with Sec. 3 describing the set up of the test cases in detail. The results and

discussion of the comparisons can be found in Sec. 4 with final conclusions in Sec. 5.



## 2 Model Descriptions

Generally, modern RTMs include a variety of tools to aid in specifying the atmospheric state and the viewing geometry. These could be relatively simple things such as pre-computed climatologies of pressure, temperature, and ozone, or something more involved such as built-in Mie scattering code to calculate the optical properties of stratospheric aerosol particles of a given size

distribution. However, the core purpose of every RTM is to solve the radiative transfer equation. Some models may contain several algorithms to do this and each one is called a *solver* or *engine*. In many cases, these solvers start from fundamentally different assumptions and have their own characteristic features. For example, the RTM SCIATRAN contains several solvers, however only one (Discrete Ordinates Method-Vector) is capable of simulating polarised radiances (Stokes vectors). Two engines are included in this study for SASKTRAN, SASKTRAN-HR and SASKTRAN-MC, which solve the radiative transfer

problem in a successive orders and Monte Carlo (MC) methods respectively.

RTMs typically belong to one of two classes: statistical or deterministic. Statistical models solve the radiative transfer equation using Monte Carlo simulation of photon paths through the atmosphere, while deterministic models use discretisation, interpolation, and various simplifying assumptions. Statistical models are often easier to implement since less assumptions are made, however they usually are orders of magnitude slower computationally.

### 2.1 Deterministic Models

Deterministic models solve the Radiative Transfer Equation (RTE) using some form of numerical integration over the Line Of Sight (LOS) and by making various simplifying assumptions. The choice of how and which quantities are discretised can result in completely different methods being applied to solving the RTE.

These methods can further be classified according to how the sphericity of the atmosphere is handled when calculating the

multiple scattered radiance field. Plane parallel models assume a flat Earth, and can therefore not be applied to simulate the limb viewing geometry. Pseudo-spherical models employ a plane parallel solution, but initialise the data in the RTE with the solar irradiance attenuated through a spherical atmosphere. Approximative spherical models trace the observer LOS through a spherical atmosphere, calculate the single scatter term spherically, and then use an approximately spherical multiple scatter source function (typically from one or more pseudo-spherical calculations, although the exact method may vary from model to

model). Lastly, fully spherical models account for sphericity in all aspects of the calculation.

Some of these approximations have been shown to have significant, systematic effects on calculated radiances in the limb viewing geometry. Most notably, approximate spherical methods which use a single plane parallel solution have been shown to be systematically high (on the order of 5%) at higher tangent altitudes (McLinden and Bourassa, 2010). Similar differences were noted by Loughman et al. (2004) in comparing approximate spherical models with fully spherical statistical models.

#### 2.1.1 GSLS

The Gauss-Seidel Limb Scattering (GSLS) RTM builds upon the techniques described by Herman et al. (1994, 1995) to simulate the vectorial radiance in a spherical atmosphere. Line of sight rays are traced through a fully spherical atmosphere,





integrating a fully spherical single scatter source function and an approximate multiple scatter source. The approximate multiple scatter source is calculated at a selected number of solar zenith angles using a pseudo-spherical calculation. The number of solar zenith angles the multiple scatter source function is calculated at depends on the solar geometry and is shown in Loughman et al. (2015). GSLS has support for atmospheric refraction and analytic computation of approximate weighting functions. A full description of GSLS can be found within Loughman et al. (2004, 2015).

GSLS has been used in several projects involving the retrieval of atmospheric constituents from limb scatter measurements. Most notably, GSLS is currently used as the RTM for the operational version of the OMPS-LP ozone and stratospheric aerosol data products (Rault and Loughman, 2013). The OMPS-LP stratospheric aerosol algorithm has also been applied to SCIA-MACHY measurements (Taha et al., 2011). GSLS was also used in experimental retrievals using limb scatter measurements from SAGE III-M (Rault, 2005).

### 2.1.2 SASKTRAN-HR

SASKTRAN is a fully spherical vector RTM originally developed at the University of Saskatchewan to process data from the Optical Spectrograph and InfraRed Imaging System (OSIRIS, Llewellyn et al., 2004) instrument. A full description of SASKTRAN can be found in Bourassa et al. (2008) and Zawada et al. (2015), and details on the polarised calculation in Dueck et al. (2017). SASKTRAN includes both a statistical (SASKTRAN-MC, see Sec. 2.2.2) and a deterministic method (SASKTRAN-HR) to solve the RTE. The deterministic approach employs the successive order of scattering technique. The technique has a physical interpretation where radiance incident from the sun directly is used to calculate the single scattered radiance. The single scatter radiance is then used to calculate the second order of scatter, and the process is iterated until convergence. Refractive effects can also optionally be included, approximate analytic weighting functions calculated, and two- and three-dimensional atmospheres can be handled. The primary application of SASKTRAN has been as the forward model for retrievals of ozone (Degenstein et al., 2009), stratospheric aerosols (Bourassa et al., 2007), and nitrogen dioxide (Sioris et al., 2017) from the OSIRIS instrument. However, SASKTRAN has also been used in a variety of projects unrelated to OSIRIS. SASKTRAN has been adapted to process limb retrievals from other instruments, including stratospheric aerosols from SCIAMACHY measurements (Rieger et al., 2018) and stratospheric ozone from OMPS-LP measurements (Zawada et al., 2018). SASKTRAN was also used to analyse data from an acousto-optical tuneable filter based instrument, the Aerosol Limb Imager (ALI, Elash et al. (2016)), which is conceptually similar to the VIS-NIR channel of ALTIUS.

SASKTRAN-HR has various options that control the accuracy of the solution, but the main one is the number of diffuse profiles, i.e. the number of discretizations used in solar zenith angle to compute the multiple scatter field. The model has been configured to use the number of diffuse profiles required to obtain approximately $0.2\,\%$ accuracy as a function of solar conditions shown in Zawada et al. (2015).

### 2.1.3 SCIATRAN

The SCIATRAN software package provides tools for modelling radiative transfer processes in the ultraviolet to thermal infrared spectral range. A detailed review of available algorithms, selected comparisons, and applications is given by Rozanov et al.





(2014). SCIATRAN contains databases (or code modules) of optical properties and climatologies, and a set of engines to solve the RTE. The only solver capable of simulating the vector radiance field is the Discrete Ordinates Method-Vector (DOM-V) solver. The DOM-V solver uses the discrete ordinates method to simulate the pseudo-spherical solution used to initialize the spherical integration.

In the limb viewing geometry SCIATRAN can operate in two modes, fully spherical and approximate spherical. In the approximate spherical mode, the single scatter radiance is calculated accounting for the full sphericity of the Earth, while the multiple scattered signal is approximated by several pseudo-spherical calculations. In fully spherical mode, the approximate spherical solution is iterated in a fully spherical geometry to account for sphericity effects. The comparisons shown use the fully spherical mode of SCIATRAN.

SCIATRAN accounts for refractive effects and calculates approximate weighting functions. SCIATRAN has been used in numerous applications spanning multiple research areas. One of SCIATRANs primary applications is its use as the forward model for SCIAMACHY limb scatter retrievals of including, but not limited to, ozone (Rozanov et al., 2007), water vapour (Rozanov et al., 2011b), and stratospheric aerosols (Von Savigny et al., 2015; Malinina et al., 2018). SCIATRAN has also been successfully used in the inversion of data products from the observations of other limb missions, including the retrievals of

stratospheric aerosol from OSIRIS measurements (Rieger et al., 2018), and ozone and stratospheric aerosol from OMPS-LP measurements (Arosio et al., 2018; Malinina et al., 2020).

### 2.2    Statistical Models

Statistical models use Monte Carlo (MC) simulation to solve the RTE. In spherical geometry, the most common technique is the so-called backward MC method, or adjoint method. Here, photons are traced, starting at the sensor, through the atmosphere

and towards the sun; this in contrast to the forward method, where photons originate at the source (Sun). Along the photon path, the choice of where the photon scatters and the direction of scattering are sampled based on the probability of a scatter event occurring. The final radiance, and associated precision, are estimated by analysing an ensemble of a large number of photons.

    The MC technique naturally has few assumptions which allows for easier implementation of new features. A primary exam-

ple of this is implementing atmospheric constituents that vary in three full dimensions, rather than only in altitude. It is also quite natural to handle the full sphericity of the atmosphere. Because of these reasons, statistical models are primarily used as benchmark models and to study new effects. Statistical models are typically orders of magnitude slower than deterministic models and are thus usually not used in operational retrieval methods. They also contain inherent random noise, driven by the number of photons used in the simulation, which may need to be considered depending on the application.

### 165   2.2.1    MYSTIC

The Monte carlo code for the phYSically correct Tracing of photons In Cloudy atmospheres (MYSTIC) model (Mayer, 2009; Emde et al., 2010) is a statistical, fully spherical, polarised RTM that is distributed as part of the libRadtran software package (Emde et al., 2016; Mayer and Kylling, 2005). Mystic is capable of simulating radiances, irradiances, heating rates, and actinic





fluxes in the solar and thermal spectral ranges. While MYSTIC was originally designed for applications in three-dimensional

cloudy atmospheres, it contains the full functionality necessary to simulate limb scattered radiances. In spherical mode, MYS-TIC uses the backward MC method. MYSTIC contains specialised variance reduction methods for handling strongly peaked phase functions (Buras and Mayer, 2011), and is capable of simulating the effects of a fully three-dimensional atmosphere. (Emde and Mayer, 2007; Emde et al., 2017). High-spectral resolution radiances can be simulated efficiently using the ALIS (Absorption Lines Importance Sampling) method (Emde et al., 2011).

### 175 2.2.2 SASKTRAN-MC

The SASKTRAN RTM contains a MC mode (SASKTRAN-MC) based upon the Siro algorithm (Oikarinen et al., 1999). The primary purpose of SASKTRAN-MC is to serve as a benchmark for the SASKTRAN-HR model.

SASKTRAN-MC is fully polarised, spherical, and also uses the backward MC method. SASKTRAN-MC is capable of handling three-dimensional atmospheres and includes options to simulate the radiance to a specific precision level, rather than

specifying the absolute number of photons to simulate. For more details see Zawada et al. (2015) and Dueck et al. (2017).

### 2.2.3 Siro

Siro is a statistical, fully spherical, polarised, RTM developed at the Finnish Meteorological Institute, using the backward MC method (Oikarinen et al., 1999). Siro is capable of simulating radiances where the atmosphere varies three-dimensionally (not only in altitude). Siro is commonly used as a reference model for both studying limb scattered radiance and in comparisons with

other RTMs. Oikarinen et al. (1999) used Siro to demonstrate the importance of multiple scattering for limb scatter instruments, and to simulate the effects of a three-dimensionally varying reflective surface (Oikarinen, 2002). In Oikarinen (2001) the effect of polarization on limb scatter radiance was assessed in detail using Siro. Siro played a key role in the RTM inter-comparison study performed by Loughman et al. (2004) as one of the MC reference models.

### 2.2.4 SMART-G

SMART-G (Speed-up Monte-carlo Advanced Radiative Transfer code with GPU) is a radiative transfer solver for the coupled ocean-atmosphere system with a wavy interface (Ramon et al., 2019) or any surface spectral BRDF boundary condition. It is based on the MC technique, works in either plane-parallel or spherical-shell geometry, and accounts for polarization. The vector code is written in CUDA (Compute Unified Device Architecture) and runs on GPUs (Graphic Processing Units). Physical processes included in the current version of the code are elastic scattering, absorption, reflection, thermal emission,

and refraction.

The radiances at any level of the domain can be estimated using the local estimate variance reduction method (Marchuk et al., 2013). Benchmark values are accurately reproduced for clear (Natraj and Hovenier, 2012) and cloudy atmospheres (Kokhanovsky et al., 2010) over a wavy reflecting surface and a black ocean (Emde et al., 2015). For pure Rayleigh atmospheres as in ocean-surface-atmosphere systems comparisons, the agreement is better than 1E-5 in intensity and 0.1% in





degree of polarization (Ramon et al., 2019; Chowdhary et al., 2020). The SMART-G code is capable of handling horizontal
inhomogeneities of the albedo like adjacency effects (Chowdhary et al., 2019), or three-dimensional variations of the oceanic
and atmospheric optical properties.

### 2.2.5   Differences in the Monte Carlo Methods

While all four models listed above use the Monte Carlo technique, there is one difference that can be noticed in the subsequent
comparisons. One option is to trace rays through the atmosphere, and calculate the scattering probability at each layer interface.
This gives the possibility of photons not scattering and directly escaping the atmosphere, which is important for estimates of
radiative fluxes (not directly applicable for limb scatter measurements). A consequence of this is that at higher wavelengths and
higher tangent altitudes where the atmosphere is optically thin, a large number of photons are required to reduce the statistical
noise to acceptable levels. In this study this technique is used by MYSTIC and SMART-G.

An alternative technique is to force every photon traced backwards from the observer to scatter. Random numbers are
generated to determine the scatter location, not if scattering actually occurs. Photons can then be weighted by the optical
thickness to account for the probability of scattering. A benefit of this technique is that the number of photons required to hit
a desired noise floor is more uniform in wavelength and altitude space, but the technique is more specific to limb scattering
measurements. Siro and SASKTRAN-MC both use the same force-scatter technique.

SMART-G includes an option to force additional scattering in limb mode, but only for the first (single) scatter. The option
has a similar effect to the technique used by Siro/SASKTRAN-MC in that it reduces the variance of the calculation for optically
thin scenarios, but it is not exactly equivalent.

Overall, all of the mentioned techniques solve the radiative transfer equation with the same level of accuracy. The only dif-
ference is the number of photons required to reach a desired level of precision. A more in-depth discussion of the computational
efficiency of the different techniques for different scenarios is presented in Sec. 4.5.

## 3   Model Test Cases

Test cases are designed to explore the aspects of the RTMs that are applicable for past, present, and future satellite-based limb-
scattering measurements. All tests are performed for the following range of tangent altitudes, solar angles, surface reflectance,
atmospheric constituent conditions, and wavelengths:

– 80 tangent altitudes from $0.5\,\mathrm{km}$ to $79.5\,\mathrm{km}$ (inclusive) with a spacing of $1\,\mathrm{km}$.

– 9 combinations of SZA and SAA which are given in Table 1.

– 3 values of a Lambertian Effective Reflectance of 0, 0.3, and 1.

– 3 atmospheric constituent conditions: pure Rayleigh scattering, Rayleigh scattering and ozone absorption, and Rayleigh
+ stratospheric aerosol scattering and ozone absorption



– 11 wavelengths provided in Table 2.

These test cases span the reasonable conditions that have been, or are currently, in use by operational limb scatter instruments
for retrievals of typical atmospheric constituents.

In addition to different atmospheric and geometry conditions, test cases are selected using different RTM settings:

   – Single scattering, vector, no refraction

– Multiple scattering, scalar, no refraction

   – Multiple scattering, vector, refraction

Note that a single scattering scalar test case would be redundant as polarization only affects $I$ through multiple scattering when
the incident source is unpolarized. In all cases the SZA and SAA are defined at the tangent point of each individual line of
sight. The placement of the tangent point assumes straight-line, un-refracted rays, from the observer.

One of the challenges in performing an inter-comparison of RTMs is ensuring that the inputs are the same across every
model. In this study, care was taken so that the input parameters were specified in a way that can be assimilated by every model
in the study. Stratospheric aerosol is specified as a log-normal distribution of Mie scattering particles with a median radius
of 80 nm and a mode width of 1.6, with scattering parameters (cross sections, Mueller matrices, and Legendre moments)
calculated using the code of Wiscombe (1980) and tabulated in wavelength. The refractive index is consistent with that of

sulfuric acid and is taken from Palmer and Williams (1975). Ozone absorption cross section is taken from measurements by
Brion et al. (1993); Daumont et al. (1992); Malicet et al. (1995) interpolated to 243 K. Rayleigh scattering is assumed to be
simple Rayleigh scattering without anisotropy corrections. These parameters are provided for reference in Table 2.

The background atmosphere is specified on a 1 km grid from 0 km to 100 km. The ozone profile is taken from a climatology
derived from measurements from the Microwave Limb Sounder (Waters et al., 2006) and the Atmospheric Chemistry Exper-

iment (Bernath et al., 2005). Rayleigh scattering number density, pressure, and temperature are taken from typical tropical
conditions in the MSIS-E-90 atmospheric model. The GLobal Space-based Stratospheric Aerosol Climatology (GLoSSAC,
Thomason et al., 2018) is used to obtain a typical background aerosol extinction.

As noted in previous polarized RTM intercomparisons (e.g. Emde et al., 2015), the Stokes vectors returned by each RTM
are not directly comparable due to differing conventions. A full discussion of differing conventions for reporting the Stokes

vector is beyond the scope of this study, and we refer to documentation on each model for specifics on how each individual
RTM defines the Stokes vector. For consistency in this study all Stokes parameters are converted to follow the definition of
Hovenier et al. (2004), which for the models included differs only by the sign of $Q$, $U$, and/or $V$. The signs applied to the
Stokes parameters from each of the models are shown in Table 3.

### 3.1 A Note on Atmospheric Gridding

The test cases specify the atmospheric state parameters on a 1 km grid from 0 km to 100 km, but leave the interpolation scheme
up to the individual RTM. The two standard choices either assume that the atmospheric state varies linearly between grid points,




**Table 1.** Solar zenith angles, solar azimuth angles, and solar scattering angles used in the test cases.

| SZA | SAA | SSA |
|-----|-----|-----|
| 10° | 20° | 80.6° |
| 15° | 70° | 84.9° |
| 35° | 90° | 90.0° |
| 45° | 50° | 63.0° |
| 50° | 130° | 119.5° |
| 60° | 70° | 72.8° |
| 70° | 30° | 35.5° |
| 80° | 60° | 60.5° |
| 80° | 150° | 148.5° |

**Table 2.** The Rayleigh scattering cross-section, ozone absorption cross-section, and stratospheric aerosol refractive index used for the test cases.

| Wavelength [nm] | Rayleigh Cross-Section [cm$^2$] | Ozone Cross-Section [cm$^2$] | Aerosol Refractive Index |
|-----|-----|-----|-----|
| 300 | 5.602831e-26 | 3.626519e-19 | $1.452272 - 0i$ |
| 315 | 4.549917e-26 | 4.218142e-20 | $1.449746 - 0i$ |
| 351 | 2.878846e-26 | 1.225798e-22 | $1.449180 - 1.33565e\text{-}05i$ |
| 435 | 1.177405e-26 | 8.362497e-23 | $1.434603 - 4.90115e\text{-}05i$ |
| 442 | 1.102286e-26 | 1.698570e-22 | $1.433675 - 5.10270e\text{-}05i$ |
| 525 | 5.438003e-27 | 2.183215e-21 | $1.429252 - 6.51460e\text{-}05i$ |
| 600 | 3.156146e-27 | 5.206045e-21 | $1.429088 - 6.46200e\text{-}05i$ |
| 675 | 1.957387e-27 | 1.505802e-21 | $1.428480 - 5.44275e\text{-}05i$ |
| 943 | 5.069933e-28 | 0 | $1.423274 - 0i$ |
| 1020 | 3.692419e-28 | 0 | $1.421113 - 0i$ |
| 1700 | 4.611621e-29 | 0 | $1.396316 - 4.33650e\text{-}04i$ |

or that the atmosphere consists of $1\,\mathrm{km}$ homogeneous layers where the atmospheric state parameters are constant. How the discretized atmospheric state maps to an effective continuous quantity is of particular importance since any retrieved quantity must be interpreted the same way.

The RTMs GSLS, SASKTRAN (HR and MC), and Siro assume linear interpolation and handle it through analytic methods. All four models calculate optical depth exactly, assuming a linearly varying extinction (see Oikarinen et al., 1999; Loughman et al., 2015, for more detail). For SASKTRAN-MC and Siro this is all that is required and the linear variation of the atmosphere is accounted for without approximation. GSLS and SASKTRAN-HR must make additional approximations in the calculation





**Table 3.** Sign that each models Stokes parameters were multiplied by to follow the definition of Hovenier et al. (2004). A "-" indicates that the component was multiplied by $-1$, while a "+" indicates the component was unchanged.

| Model | Q | U |
|---|---|---|
| GSLS | - | + |
| MYSTIC | + | - |
| SASKTRAN (MC/HR) | - | + |
| SCIATRAN | + | + |
| Siro | - | + |
| SMART-G | + | - |

of the source function and we refer back to Loughman et al. (2015) and Zawada et al. (2015) respectively for the exact methods used.

The other RTMs (MYSTIC, SCIATRAN, SMART-G) use homogeneous layers. To compare the models more directly, the RTMs using homogeneous layers have been configured using sub-layers with linear interpolation. Figure 2 shows an example of how using $1000\,\mathrm{m}$ or $250\,\mathrm{m}$ homogeneous shells vary compared to $100\,\mathrm{m}$ shells within MYSTIC. Layering using $1000\,\mathrm{m}$ shells introduces errors on the order of $0.5\,\%$ at $351\,\mathrm{nm}$ in regions where the atmosphere is optically thin, while the error using $250\,\mathrm{m}$ shells is $0.1\,\%$. For all future calculations both MYSTIC and SMART-G have been configured to use $250\,\mathrm{m}$ shells. SCIATRAN uses a hybrid system where sub-gridding is applied only to the first three layers near the beginning of each integration line and three layers above the tangent point (the exact number of layers is an input parameter).

## 4   Discussion and Results

The main challenge in interpreting, and attributing, differences between models is that the true answer is not known. All comparisons shown in this section are relative to what we have called the Multi-Model Mean (MMM). The MMM is composed of the set of models that agree with each other to a level that cannot be attributed to a concrete difference in a single model. For the single scattering test cases the level was determined to be $0.2\,\%$, and $1\,\%$ for the multiple scattering test cases. Any RTM that is found to have disagreements above these levels is excluded from the MMM for the relevant test case, causing the models composing the MMM to vary between different test cases.

While tests were performed for both vectorial and scalar modes, no significant differences were found in agreement between the models in scalar or vector mode. Therefore, for the purpose of brevity, no comparisons of strictly scalar mode are shown. It is understood that the results for the polarised comparisons are equally applicable to scalar calculations.

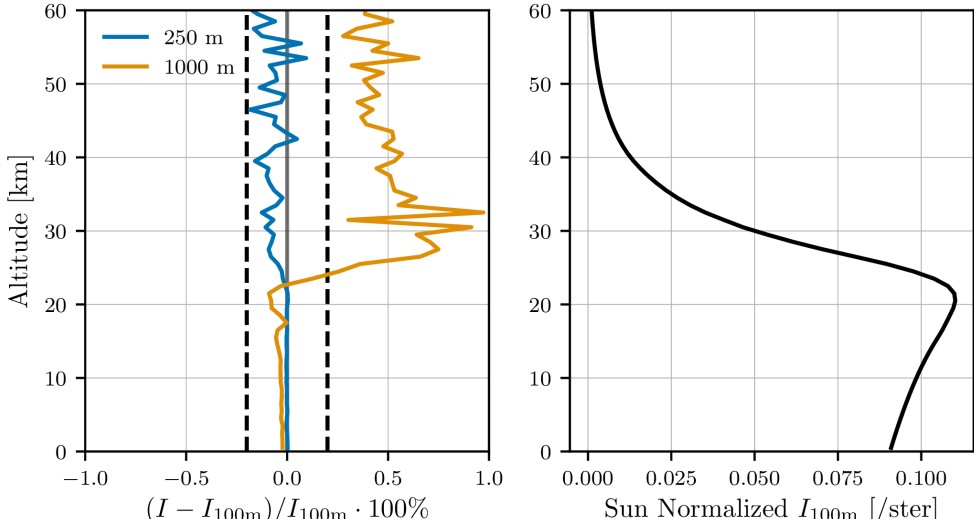

**Figure 2.** (Left Panel) Percent difference in single scattered radiance when MYSTIC is using $1000\,\mathrm{m}$ or $250\,\mathrm{m}$ homogeneous shells compared to $100\,\mathrm{m}$ homogeneous shells. (Right Panel) The sun-normalized radiance computed using $100\,\mathrm{m}$ shells. The atmosphere contains Rayleigh scattering, ozone absorption, and stratospheric aerosol Mie scattering with a SZA of $70°$ and a SAA of $30°$. Refraction is disabled. MYSTIC was configured to use 100 million photons. Dashed vertical lines indicate $\pm0.2\,\%$ levels.

## 4.1 Single Scatter

Simplifying assumptions made for the single scatter calculation are minimal, and thus we expect differences to be relatively
290  small. We do not expect zero differences due to the various methods of gridding in the vertical dimension of the atmosphere. As originally done in the limb geometry by Siro (Oikarinen et al., 1999) and discussed extensively in Loughman et al. (2004, 2015), the calculation of optical depth may be performed analytically assuming an extinction that varies linearly in altitude. However, the integration of the source function cannot be performed the same way and an approximation must be made. For example, SASKTRAN-HR creates a second order spline of the source function across an integration cell (Zawada et al., 2015),
295  but other models may assume a constant source function or do something more sophisticated. All of this is further complicated by any form of sub-gridding the model may perform in order to obtain a more accurate result.

Differences for the most extreme single scatter case (Rayleigh + ozone + stratospheric aerosol) are shown in Fig. 3. Differences are presented relative to the mean of the three deterministic models (SASKTRAN-HR, GSLS, and SCIATRAN), which we refer to as the Multi-Model Mean (MMM). We chose to use the deterministic models as the reference because statistical
300  errors in the single scatter case are on the order of the differences observed between the different models. For all conditions, errors between the three deterministic models are less than $0.1\,\%$. The differences between GSLS, SASKTRAN-HR, and SCI-ATRAN are likely due to the slightly different gridding techniques. For the statistical models MYSTIC, SASKTRAN-MC, SMART-G, and Siro no errors are detected that are greater than the random noise present ($\sim0.2\,\%$ in most cases). One thing





to note about the SMART-G calculation is that statistical errors are correlated in solar geometry, while for the other MC models the errors are uncorrelated. Errors are correlated because the SMART-G calculation considered multiple solar positions simultaneously in a single calculation, while the other models performed each line of sight and solar position independently.

The excellent agreement in single scatter is expected due to the relative simplicity of the calculation. Fundamentally each RTM solves the single scatter problem in the same way with minimal assumptions, the primary purpose of this test is to ensure that the inputs to RTM are configured correctly. The agreement of 0.1–0.2 % here sets a baseline that differences above this level in more complex test cases cannot be explained by differing treatment of input data.

## 4.2 Multiple Scatter

Differences in multiple scatter are expected to be larger than those seen in single scatter owing to the extra complexity of the radiative transfer problem. The discrete models must deal with discretizations of the multiple scattering source term and may also make fundamental approximations for the sake of computational speed. Comparatively the statistical models employ a simpler technique.

In Fig. 4 we see that differences between each RTM and the MMM (MYSTIC, SASKTRAN-HR, SASKTRAN-MC, SCIATRAN, and SMART-G) is usually less than 1 % with a few exceptions. Siro shows a low bias on the order of 1–4 % at 351 nm, which is most pronounced at low solar zenith angles. The bias is not present when the Lambertian surface albedo is set to 0 instead of 1. This bias was present in Loughman et al. (2004), however it was misattributed to a high bias in the other RTMs as Siro was used as the reference model. Internal testing suggests that the difference may not be directly related to ground scattering, and instead is an error that compounds on each succesively higher order of scatter. The error is strongest near 351 nm and a surface albedo of 1 because this is the condition where higher orders of scatter have the largest contribution to the observed radiance.

GSLS at 351 nm has a distinct pattern in altitude with variations on the order of 1 % that are relatively consistent across solar geometry. The exact cause of these variations are unknown but are likely caused by discretizations of the multiple scatter source field calculation, however as stated these variations are fairly small. At 675 nm GSLS shows significant deviations of up to 4 % depending on the solar geometry at altitudes near 25 km. The deviation is only present when the Lambertian surface albedo is 1 and is almost non-existent with a 0 albedo. Testing has shown that the difference is present for all atmospheric composition scenarios and is thought to be due to approximations made in the ground to line of sight scatter calculation.

SASKTRAN (HR and MC), SCIATRAN, SMART-G, and MYSTIC all agree for all conditions to better than 1 %. As stated, differences less than 1 % are difficult to attribute to any particular RTM due to both not having a precise reference model and the inherent statistical noise. However, we do note that SASKTRAN-HR and SCIATRAN both have altitude variation patterns on the order of 0.5 % that are constant across solar geometry but differ in wavelength. MYSTIC, SASKTRAN-MC, and SMART-G are indistinguishable at the level of statistical noise. The general agreement between the models is better than has been observed before in comparisons of RTMs in limb scatter geometry even for scalar cases.





**Figure 3.** Percent differences in single scatter $I$ relative to the MMM (GSLS, SASKTRAN-HR, and SCIATRAN) for each RTM at a variety of wavelengths and solar conditions. The atmospheric optical properties include Rayleigh scattering, ozone absorption, and stratospheric aerosol Mie scattering. Refraction is disabled. Dashed vertical lines indicate ±0.2 %



**Figure 4.** Percent differences in $I$ relative to the MMM (MYSTIC, SASKTRAN-HR, SASKTRAN-MC, SCIATRAN, and SMART-G) for each RTM at a variety of wavelengths and solar conditions. The atmospheric optical properties include Rayleigh scattering, ozone absorption, and stratospheric aerosol Mie scattering. Refraction is disabled. Dashed vertical lines indicate $\pm 1\%$ levels.





### 4.3 Stokes Parameters

While all comparisons so far have involved vector radiative transfer calculations only differences in the $I$ component of the Stokes vector have been analyzed. Agreement in $I$ is indicative that the polarization implementation in each model is sensible, however polarization can be investigated more rigorously by analyzing the individual Stokes parameters. Figure 5 shows the

individual Stokes components, and their differences to the MMM (MYSTIC, SMART-G, and SASKTRAN-MC), for one scenario with large polarization (scattering angle near $90\,^\circ$, albedo 0) and one scenario with low polarization (scattering angle away from $90\,^\circ$, albedo 1).

Generally agreement between $Q$, $U$, and the linear polarization, $\sqrt{Q^2 + U^2}$, is worse than the agreement with $I$. MYSTIC, SASKTRAN-MC, and SMART-G agree for all components at the level of statistical noise ($\sim$0.2–1%), while Siro shows

deviations at altitudes below $30\,\mathrm{km}$ that can approach $5\,\%$ depending on the condition. SASKTRAN-HR and SCIATRAN are usually within $1\,\%$ for the cases shown, however in some situations this can exceed $1\,\%$. In cases where the polarization is small, GSLS can show deviations of $2$–$4\,\%$, but these disappear in the cases with high linear polarization. The differences observed in linear polarization sometimes mirror differences observed in $I$, and are sometimes independent.

In order to isolate the polarization effects, we consider two quantities: the Degree of Linear Polarization (DOLP) and the

Linear Polarization Orientation (LPO). DOLP is defined as

$$\mathrm{DOLP} = \frac{\sqrt{Q^2 + U^2}}{I}, \qquad (1)$$

which is the fraction of radiation that is linearly polarized, and LPO is defined as

$$\mathrm{LPO} = \frac{1}{2}\arctan\frac{U}{Q}, \qquad (2)$$

which indicates the direction of linear polarization. The absolute differences in DOLP and LPO for the Rayleigh scattering,

ozone absorption, and stratospheric aerosol scattering case are shown in Fig. 6 and Fig. 7 respectively.

Differences in the DOLP and LPO are overall small, with a few exceptions. Once again, the MC models MYSTIC, SASKTRAN-MC, and SMART-G agree in all cases to the level of statistical noise in the computation. At $300\,\mathrm{nm}$ no differences are observed between the models as the majority of the signal is single scatter. GSLS, SASKTRAN-HR, and SCIATRAN all have minor spreads in DOLP at $351\,\mathrm{nm}$ depending on solar angle and albedo on the order of 0.002. SCIATRAN shows differences in LPO

at $351\,\mathrm{nm}$ of $0.2\,^\circ$ for some conditions, however these are conditions where the overall polarization signal is small. At $675\,\mathrm{nm}$ GSLS has differences in DOLP of up to 0.02 when the surface albedo is 1, which are likely related to the differences in $I$ observed previously. For the same wavelength the LPO shows deviations on the order of $0.5\,^\circ$ for the high surface albedo case.

Siro has differences in DOLP, but curiously does not have any significant differences in LPO. At $351\,\mathrm{nm}$, deviations in DOLP are up to 0.03 and are present at all albedos and solar conditions, but are larger at high albedo and low solar zenith

angle. The deviations are largest at conditions where there was significant differences in $I$, but do not share the same shape. There are differences of up to 0.02 at $675\,\mathrm{nm}$ in DOLP between Siro and the MMM, however the differences are only present at low albedos. The differences are largely eliminated when aerosol is removed from the atmosphere (not shown) suggesting that it could be due to aerosol multiple scattering.



**Figure 5.** Left most column: The MMM calculation of $I$, $Q$, and $U$ for four different scenarios consisting of MYSTIC, SASKTRAN-MC, and SMART-G. Other columns: percent difference in $I$, $Q$, $U$, and $\sqrt{Q^2 + U^2}$ for each model relative to the MMM. The atmospheric optical properties include Rayleigh scattering, ozone absorption, and aerosol scattering. Dashed vertical lines indicate $\pm 1\%$ levels.





Both SASKTRAN (HR and MC) and GSLS make the assumption that $V$ is identically 0, which reduces the size of the phase

matrix to speed up the computation, and it does not appear that this approximation affects the results in a noticeable way. The comparison atmospheres only include smaller spherical scatterers (Mie and Rayleigh scattering), and do not include larger particles as would be seen in ice clouds for example. It is possible that the approximation of neglecting $V$ would break down under conditions containing larger particles, droplets or crystals.

### 4.4 Refraction

All of the models considered thus far with the exceptions of SASKTRAN-MC and MYSTIC support atmospheric refraction to some level. While Siro has support for refraction it was not tested as part of this study. GSLS and SASKTRAN-HR neglect refraction of the incoming solar rays. Furthermore GSLS and SASKTRAN-HR neglect refraction for multiple scattering effects, only implementing refraction for the line of sight ray. SCIATRAN and SMART-G implement refraction in a generic way accounting for all solar and multiple scattering effects. SASKTRAN-HR has since been updated to include refractive effects

for incoming solar rays and multiple scatter effects but the calculations here use only line of sight refraction.

Differences in the multiple scatter signal when refraction is enabled for the stratospheric aerosol scattering case are shown in Fig. 8. At $351\,\mathrm{nm}$ the effect of refraction is minimal and agreement is identical to the cases without refraction; however, at longer wavelengths several differences are observed between the RTMs. SMART-G has a discontinuity in the signal at $11.5\,\mathrm{km}$ causing differences on the order of $1\,\%$ relative to SASKTRAN-HR and SCIATRAN. GSLS shows similar differences in the

refracted and unrefracted cases, indicating that the refractive effect is similar to that of SASKTRAN-HR and SCIATRAN.

To further investigate these differences, the ratio of refracted to unrefracted $I$ for a single condition for each model is shown in Fig. 9. This refraction ratio was found to be insensitive to the solar geometry, albedo, and atmospheric composition. The refraction ratio is larger at longer wavelengths due to the atmosphere being more optically thin, however the 1020 nm wavelength as shown in the figure is representative of the differences observed between the models at all wavelengths. GSLS,

SASKTRAN-HR, and SCIATRAN show excellent agreement in the refraction ratio, with differences being insignificant next to the already observed differences between the models. The refractive enhancement in SMART-G is $1$–$2\,\%$ less than GSLS, SASKTRAN-HR, and SCIATRAN with the exception of a discontinuity near $11.5\,\mathrm{km}$ that greatly enhances the refractive enhancement for a few km below. Possible reasons for the differences of SMART-G compared to GSLS, SASKTRAN-HR, and SCIATRAN are still under investigation.

There are several possible reasons for the small observed differences between GSLS, SASKTRAN-HR and SCIATRAN. The index of refraction of the atmosphere was not harmonized between the models, instead each model performed internal calculations using the provided atmospheric temperature and pressure. There exist various methods to do this calculation and they may not be the same between each RTM. Differing methods of ray-tracing and integration can also lead to small differences. Since SCIATRAN (and SMART-G) included refractive effects for the incoming solar ray and multiple scatter

terms, it is possible that this could be the source of some minor differences. The solar geometries tested here have been limited to SZA $\leq 80\,^\circ$, where refraction of the incoming solar ray is expected to be minimal. Further study could examine the effect of refraction at higher solar zenith angles.







**Figure 6.** Absolute differences in DOLP relative to the MMM (MYSTIC, SASKTRAN-MC, and SMART-G) for each RTM at a variety of wavelengths and solar conditions. The atmospheric optical properties include Rayleigh scattering, ozone absorption, and stratospheric aerosol Mie scattering. Refraction is disabled. Dashed vertical lines indicate ±0.005 levels.





**Figure 7.** Absolute differences in LPO [°] relative to the MMM (MYSTIC, SASKTRAN-MC, and SMART-G) for each RTM at a variety of wavelengths and solar conditions. The atmospheric optical properties include Rayleigh scattering, ozone absorption, and stratospheric aerosol Mie scattering. Refraction is disabled. Dashed vertical lines indicate ±0.2° levels.





**Figure 8.** Same as Fig. 4 but with refraction enabled. The MMM is composed of SASKTRAN and SCIATRAN. Dashed vertical lines indicate ±1% levels.





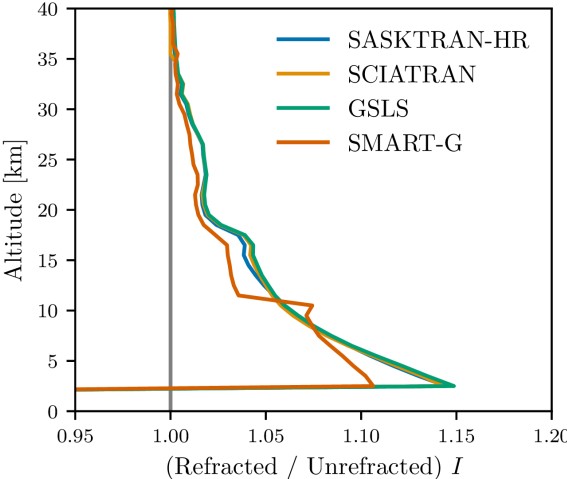

**Figure 9.** Ratio of refracted to unrefracted $I$ for multiple scatter, Rayleigh scattering + ozone absorption + stratospheric aerosol scattering, SZA=70°, SAA=30°, and a Lambertian surface reflectance of 0 at 1020 nm for every model that supports refraction.

## 4.5 Timing

We have considered a basic run time comparison between the models in the study. While a useful exercise, there are technical challenges in standardizing the hardware used to execute the models, and in the case of SMART-G which uses a GPU the standardization is not possible. More importantly every model has settings that involve an accuracy/speed trade-off. For example, with the deterministic models, there are various discretization setting that can dramatically affect the speed of the calculations.

Harmonizing the balance between accuracy and speed between all of the RTMs is impractical, instead we opt for a simple order of magnitude timing estimate. The time taken to execute all of the multiple scatter polarized tests for each model without refraction is shown in Table 4. These timing numbers should be interpreted as the time required to execute a wide variety of test cases, individual RTMs may be significantly more or less efficient in specific cases, however analyzing these differences is beyond the scope of this study. For the CPU based models a scaled runtime value is also supplied where the runtime on equivalent hardware has been approximated using the relative CPU benchmark values from https://www.passmark.com/. The deterministic models, GSLS, SCIATRAN, and SASKTRAN-HR, all have runtimes of a similar order of magnitude taking anywhere from 0.3 s to 1 s to execute a single wavelength, solar geometry, and atmospheric composition on average.

Analyzing the timing of the Monte Carlo models is inherently more challenging as the calculations also contain statistical noise. It is common to benchmark models for a set number of photons, however the number of photons used is not comparable between Siro/SASKTRAN-MC and MYSTIC/SMART-G since the MC technique is not the same. Instead, the precision of the calculation must be directly compared, which is shown for a typical condition in Fig. 10. For the precision estimation and timing, MYSTIC was configured to use a constant 1E6 photons. The other models (SASKTRAN-MC, Siro, and SMART-G)





**Table 4.** Estimated time to execute all multiple scatter no refraction tests. This includes three effective surface albedos, three different atmospheric compositions, 11 wavelengths, 9 solar geometries, and 80 lines of sight. The MC models were executed to the precision shown in Fig. 10 (see text for more detail). $^a$The scaled runtime is calculated by scaling every CPU to the computational power of the AMD 3900x using the relative benchmark values from https://www.passmark.com/ as of November 6th 2020.

| Model | Hardware Description | Time [minutes] | Scaled Runtime$^a$ [Minutes] |
|---|---|---|---|
| GSLS | Two Intel Xeon E5-2630 (6 physical cores at 2.3 GHz each) | 35 | 13.0 |
| SASKTRAN-HR | AMD 3900x (12 physical cores at 3.8 GHz) | 7.4 | 7.4 |
| SCIATRAN | Intel i7-6850 (6 physical cores at 3.6 GHz) | 13.5 | 4.6 |
| SASKTRAN-MC | AMD 3900x (12 physical cores at 3.8 GHz) | 1909 | 1909 |
| Siro | Four Intel Xeon E5-2630 (8 physical cores at 2.4 GHz each) | 16560 | 20078 |
| MYSTIC | AMD 3900x (12 physical cores at 3.8 GHz) | 1906 | 1906 |
| SMART-G | NVIDIA Titan V | 59.2 | N/A |

were configured identically to the previous radiance comparisons. Siro used a constant 1E6 photons, while SASKTRAN-MC and SMART-G used a variable number of photons targeting 0.2–0.3% precision.

For the given precisions, The CPU based statistical models, MYSTIC, SASKTRAN-MC, and Siro, have runtimes within an order of magnitude. The runtime of Siro appears large, however the precision is generally better. Approximately scaling the
Siro calculation to 0.2% precision would result in a speed increase of a factor of ~4. Because of the differences in precision between the different calculations we won't attempt to quantify small differences between the MC models. One thing of particular interest is the general efficiency between the technique used by Siro/SASKTRAN-MC and MYSTIC/SMART-G. Both Siro and MYSTIC used a constant 1E6 photons for all conditions, however the precision of Siro is relatively constant in altitude and wavelength hovering around 0.1%, while the precision of MYSTIC varies significantly. MYSTIC achieved better
than 0.1% precision in cases where the atmosphere heavily scatters (low altitudes, shorter wavelengths), and worse precision at higher altitudes and longer wavelengths where the atmosphere is optically thin. As mentioned earlier, this is since photons in Siro are forced to scatter, while photons in MYSTIC may pass through the atmosphere without interaction.

The runtime for the GPU based MC model, SMART-G, is ~1–2 orders of magnitude less than the other MC models. The most natural comparison is between SMART-G and SASKTRAN-MC since they have similar precision for this case. Here,
SMART-G achieves a speedup of ~30x relative to SASKTRAN-MC on the hardware used. SMART-G only forces scatter events to happen on the first order of scatter (contrary to SASKTRAN-MC and Siro which force scatters on all orders), but it appears that this is sufficient to obtain reasonable precision in all scenarios.

## 5 Conclusions

A systematic comparison has been performed between seven radiative transfer models operating in the limb scatter geometry.
The seven models are capable of handling the sphericity of the atmosphere, and compute the Stokes vector accounting for





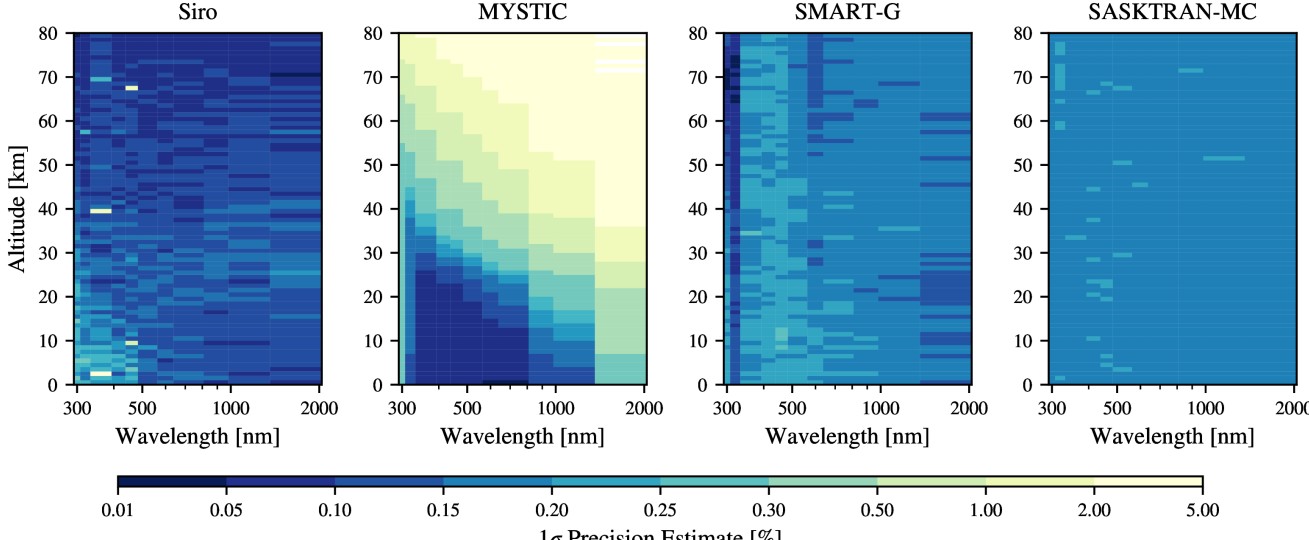

**Figure 10.** Precision estimates for the MC models for multiple scatter, Rayleigh scattering + ozone absorption + stratospheric aerosol scattering, SZA=70°, SAA=30°, and a Lambertian surface reflectance of 1. Precision estimates for MYSTIC, SASKTRAN-MC, and SMART-G were taken from the model output. Siro precision was estimated by running the above scenario 20 times and taking the standard deviation.

polarisation. The test cases cover a wide variety of solar angles, Rayleigh scattering, ozone absorption, Mie scattering, and surface reflectances.

In single scatter, the deterministic models GSLS, SASKTRAN-HR, and SCIATRAN agree within 0.1 % for all observed conditions. The statistical models MYSTIC, SASKTRAN-MC, Siro, and SMART-G all agree to a level better than the precision of the calculation which is approximately ∼0.2 %.

For almost all conditions in multiple scatter, the agreement between the fully spherical models in the multiple scatter cases is within 1% when refraction is disabled for $I$ with a few exceptions:

- Siro can have disagreement of up to 3 % at shorter wavelengths, particularly when the solar zenith angle is low and the surface reflectance is high. The difference manifests as a low bias in the radiance, and a high bias in the degree of linear polarization. The cause of this bias is currently unknown.

- At longer wavelengths in all atmospheric conditions and when the Lambertian surface reflectance is high, GSLS shows biases of up to 3 % that are dependent on solar geometry. The bias is thought to be caused by approximations made in the direct ground to line of sight scattering calculation.

Refraction has been tested for GSLS, SASKTRAN-HR, SMART-G, and SCIATRAN. The refractive effect among all models is almost indistinguishable, with the exception of a ∼1 % discontinunity in radiance in the SMART-G calculation at 11.5 km when refraction is enabled. The cause of the discontinuity is currently unknown.



Differences in quantities representing linear polarization, the DOLP and LPO, have also been assessed, however the results are more difficult to interpret. The MC models MYSTIC, SASKTRAN-MC, and SMART-G agree within statistical noise for all considered conditions and serve as a combined reference. SASKTRAN-HR and SCIATRAN generally agree with the reference at a level of 0.002 in DOLP and 0.2° in LPO, with the largest deviations in LPO being in conditions where the linearly polarized signal is small. For most conditions, GSLS agrees at a similar level, with the exception of high Lambertian surface albedos at longer wavelengths where DOLP can vary up to 0.02, and LPO by 0.5°. Siro shows deviations in DOLP at both 351 nm and 675 nm that approach 0.03, but has no distinguishable difference from the reference in LPO.

Overall the agreement between the models is excellent, and is better than has been reported for scalar comparisons in the past. The agreement provides additional confidence in the retrievals from limb scatter instruments such as OMPS-LP, OSIRIS, and SCIAMACHY. In particular, confidence in modelling the polarized signal is important for the upcoming ALTIUS mission which is linearly polarized.

There are several areas where future studies comparing RTMs in the limb viewing geometry could expand upon. Scattering from larger, non-spherical, particles, droplets, or crystals should be assessed which may result in larger differences in particular for circular polarization. More extreme cases with larger solar zenith angles may be checked, in particular to determine the effect of refraction at higher solar zenith angles.

*Data availability.*  The input atmospheric data, test cases, and the results for each model are made publicly available as Zawada et al. (2020).

*Author contributions.*  D.Z. wrote the initial draft of the manuscript. Model runs were performed by D.Z., G.F., R.L., A.M., and A.R.. All authors contributed in designing the study, interpreting the results, and revising the manuscript.

*Competing interests.*  No competing interests are present.

*Acknowledgements.*  The work has been partially supported by the Canadian Space Agency, the European Space Agency, and the state and the university of Bremen. Advancements of the radiative transfer model SCIATRAN made are a contribution to the project VolARC funded by the German Research Foundation (DFG) through the research unit VolImpact (FOR2820). The work by A.M. was supported by Academy of Finland Centre of Excellence in Inverse Modelling and Imaging (project number 312125). The work of R.L. was funded by NASA contract 80NSSC18K0847 (led by Ghassan Taha), and R.L. appreciates the support of Ghassan Taha and Tong Zhu for the code testing work done for this project. Dan Kahn, Jason Li, Mike Linda, and Colin Seftor also provided valuable assistance with NASA computer access, code setup and timing assessments.



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
