# Peer review of "Systematic Comparison of Vectorial Spherical Radiative Transfer Models in Limb Scattering Geometry"

_Atmospheric Measurement Techniques, 2020_

## Referee Comment (RC1)

Even though this is a difficult topic, the paper is very well written and understandable. I recommend it for publication after the following minor corrections. This long paper could be made shorter by dropping the timing section. I didn't find it that interesting, particularly since apples-to-apples comparisons were difficult, particularly for the Monte Carlo models. Perhaps that section could be condensed, if it is not eliminated.

L36: BrO reference should not be McLinden and Bourassa, 2010. It should be McLinden et al.,2010:

C. A. McLinden, C. S. Haley, N. D. Lloyd, F. Hendrick, A. Rozanov, B.-M. Sinnhuber, F. Goutail, D. A. Degenstein, E. J. Llewellyn, C. E. Sioris, M. Van Roozendael, J. P. Pommereau, W. Lotz, J. P. Burrows, Odin/OSIRIS observations of stratospheric BrO: Retrieval methodology, climatology, and inferred $Br_y$, J. Geophys. Res., **115**, D15308, doi:10.1029/2009JD012488 (2010).

L48: "at most" -> ", at most,"

L75: "built-in" -> "a built-in"

L79: With the previous sentence mentioning polarization, it should be clear here whether Sasktran HR is a vector model or not.

L105: "the multiple scatter source function is calculated at" ->"at which the multiple scatter source function is calculated"

L106: In what sense are the weighting functions "approximate"? Is this related to the pseudo-spherical multiple scattering?

L140 (and elsewhere): approximate -> approximately

L146: SCIATRANs -> SCIATRAN's

L172: The last part of this sentence is repetitive: "and is capable of simulating the effects of a fully three-dimensional atmosphere".

L180 (and elsewhere): A comma should follow a leading prepositional phrase. See L76, L77 for good examples.

L186: Was the surface 3-d (i.e. varying terrain elevation) or is 2-d meant?

L206: The two sentences starting here are irrelevant to this paper. Maybe they should be deleted.

L207: "higher wavelengths" -> "longer wavelengths" ?

L214: "force-scatter" What is this? Never heard of such a term. It becomes clear later (L432), but I suggest a rewording here.

L237 Reword or remove "when the incident source is unpolarized". This is not correct. The incident source could be, for example, partially polarized and the statement would be true.

L245: Begin sentence with "The ozone…"

L247: "simple Rayleigh scattering without" -> "elastic and without"

L279: Remove comma after "attributing"

L288 (Figure 2 caption): State the wavelength. Presumably there is a single wavelength used to generate the figure.

L299: "(MMM)." -> "(MMM) for this case."        [see L284]

L304: in -> with                                (?)

L306: "simulation" might be preferable over "calculation"

L329: What kind of approximations are made in the ground-to-LOS scatter? How might GSLS be approximating this differently? Since the surface is Lambertian for all models, it does not seem that this should be a source of bias.

L369: identically -> exactly

L384: "differences" -> "differences relative to SMART-G"          (?)

L388: "The refraction ratio is larger at longer wavelengths due to the atmosphere being more optically thin". This explanation is insufficient for me. Is it that refraction is of greater relative importance when scattering is diminished? If so, I think my wording is more to the point.

L397: Odd construction with this sentence (suggested change is optional):  "There exist various methods" -> "Various methods exist"

L400: The meaning of "this" is ambiguous. Can you not narrow it down with some single scattering comparisons?

L402 (and L471): higher -> larger  (see L470)

L415: Delete ", solar geometry, and atmospheric composition"

L423: The -> the

L441: Search the document for polaris* and replace the 8 occurrences with polariz*

L448: low -> small

L455: Is refraction considered at all altitudes for SMART-G or does it "turn on" when the altitude is low enough (e.g. 11.5 km)?

L477: university -> University

L720: No need to provide second website and publisher in this reference and many others, or is this a new convention?

---

## Author Comment (AC2)

**Responses to Referee 1 (Christopher Sioris) on Behalf of the Authors**

We would like to thank the referee for their helpful comments and suggestions. Included below is each of the referee's comments (italics) followed by our reply.

**General Comments**
* * *
*Even though this is a difficult topic, the paper is very well written and understandable. I recommend it for publication after the following minor corrections. This long paper could be made shorter by dropping the timing section. I didn't find it that interesting, particularly since apples-to-apples comparisons were difficult, particularly for the Monte Carlo models. Perhaps that section could be condensed, if it is not eliminated.*

**Reply**: We thank the reviewer for the positive comments on the paper. The point about the timing section does not fall on deaf ears, however we think it does add value to the manuscript. Having the timings presented indicates to the reader that every model in the study used "reasonable" settings, i.e., no RTM was executed with settings that could not be used in practice. The timing also highlights a few of the practical differences between the different MC techniques used by SASKTRAN-MC/Siro and SMART-G/MYSTIC, and demonstrates an important result in that the "forced single scatter only" method of SMART-G is very efficient. The referee's point that this is not an "apples-to-apples" comparison is very valid, and this is the reason why the section may be considered verbose in a few places. We wanted to be clear and precise in how the results should and should not be interpreted which makes it difficult to shorten the section.

**Specific Comments**
* * *
**L36** *BrO reference should not be McLinden and Bourassa, 2010. It should be McLinden et al.,2010*

**Reply**: Thank you, this has been fixed.
* * *
**L48** *"at most" → ", at most,"*

**Reply**: Changed.
* * *
**L75** *"built-in" → "a built-in"*

**Reply**: Changed.
* * *
**L79** *With the previous sentence mentioning polarization, it should be clear here whether Sasktran HR is a vector model or not.*

**Reply**: Good point, we have added the descriptor "polarized".
* * *
**L105** *"the multiple scatter source function is calculated at" → "at which the multiple scatter source function is calculated"*

**Reply**: Added.
* * *
**L106** *In what sense are the weighting functions "approximate"? Is this related to the pseudo-spherical multiple scattering?*

**Reply**: The weighting functions are "approximate" in that the contribution from single scatter terms is handled exactly, but the contribution from multiple scatter has to be handled approximately. This is not related to how well the RTM itself calculates the multiple scattering solution, it is something specific to the weighting functions and more of a computational approximation. Since weighting functions are not a focus of the paper we think it is okay to the leave the wording as is since the next line refers to two other papers where

more information can be found.
* * *
***L140 (and elsewhere)*** *approximate → approximately*

**Reply**: We see the grammar point, but "approximate spherical" has become the standard term for describing this type of solution so we prefer to leave it.
* * *
***L146*** *SCIATRANs → SCIATRAN's*

**Reply**: Changed.
* * *
***L172*** *The last part of this sentence is repetitive: "and is capable of simulating the effects of a fully three-dimensional atmosphere".*

**Reply**: We have changed this to read "and is capable of handling atmospheres where the parameters vary in three-dimensions (not just in altitude)".
* * *
***L180 (and elsewhere)*** *A comma should follow a leading prepositional phrase. See L76, L77 for good examples.*

**Reply**: Added .
* * *
***L186*** *Was the surface 3-d (i.e. varying terrain elevation) or is 2-d meant?*

**Reply**: You are correct, 2-D is meant and this has been changed.
* * *
***L206*** *The two sentences starting here are irrelevant to this paper. Maybe they should be deleted.*

**Reply**: We understand the referees point that the distinction between the two different MC techniques is not particularly relevant for the presented radiance comparisons, however, the

difference is very important for the timing section.
* * *
**L207** *"higher wavelengths"* → *"longer wavelengths" ?*

**Reply**: Thank you, longer is better.
* * *
**L214** *"force-scatter" What is this? Never heard of such a term. It becomes clear later (L432), but I suggest a rewording here.*

**Reply**: This has been reworded to "Siro and SASKTRAN-MC both use the same technique where every photon traced is forced to scatter"
* * *
**L237** *Reword or remove "when the incident source is unpolarized". This is not correct. The incident source could be, for example, partially polarized and the statement would be true.*

**Reply**: We have changed the statement "polarization only affects $I$ through multiple scattering when the incident source is unpolarized." to read "the single scatter $I$ is unaffected by polarization when the incident source is unpolarized". The intended message is that in the scattering plane, $I_{out} \sim P_{11}I_{in} + P_{12}Q_{in}$, and if the incident source is unpolarized then $Q_{in}$ is 0 thus the output radiance is unaffected by polariation. This would also be true for a circularly polarized input source, but here we aren't claiming it is a necessary condition, only a sufficient condition.
* * *
**L245** *Begin sentence with "The ozone..."*

**Reply**: Changed.
* * *
**L247** *"simple Rayleigh scattering without"* → *"elastic and without"*

**Reply**: Changed.
* * *
**L279** *Remove comma after "attributing"*

**Reply**: Removed.
* * *
**L288 (Figure 2 caption)** *State the wavelength. Presumably there is a single wavelength*

*used to generate the figure.*

**Reply**: Added.
* * *
***L299*** *"(MMM)." → "(MMM) for this case." [see L284]*

**Reply**: Changed.
* * *
***L304*** *in → with (?)*

**Reply**: Changed.
* * *
***L306*** *"simulation" might be preferable over "calculation"*

**Reply**: We agree.
* * *
***L329*** *What kind of approximations are made in the ground-to-LOS scatter? How might GSLS be approximating this differently? Since the surface is Lambertian for all models, it does not seem that this should be a source of bias.*

**Reply**: We think our wording here was a little confusing, it is not an approximation in the ground scatter itself since as you point out it should be fairly simple, instead it is more of an approximation in the full process of accounting for ground multiple scattering. The current thought is that it involves
 All that being said, this is only the current theory and it needs further investigation, which is why we chose not provide additional details. When we mention this (once in the main text and once in the conclusions) we have made sure to state that it is part of the multiple scattering calculation and still under investigation.
* * *
***L369*** *identically → exactly*

**Reply**: Changed.
* * *
***L384*** *"differences" → "differences relative to SMART-G" (?)*

**Reply**: We have reworded this entire phrase to be more clear, "The agreement of GSLS

relative to the other models is almost identical in the refracted and unrefracted cases"
* * *
**L388** *"The refraction ratio is larger at longer wavelengths due to the atmosphere being more optically thin". This explanation is insufficient for me. Is it that refraction is of greater relative importance when scattering is diminished? If so, I think my wording is more to the point.*

**Reply**: Basically at short wavelengths and low altitudes, there is so much scattering (or absorption) that you don't "see" low altitudes where refraction is actually important. So you are correct that the diminished scattering leads to refraction being more important, but it is not the sole reason. We have split this sentence into two and it now reads "At short wavelengths and low tangent altitudes, the increased extinction causes the atmosphere to be optically thick, reducing the contribution from the lower atmospheric layers where the refractive effects are significant. Therefore the refraction ratio is shown at 1020 nm which is representative of the differences observed between the models at all wavelengths where the atmosphere is optically thin."
* * *
**L397** *Odd construction with this sentence (suggested change is optional): "There exist various methods" → "Various methods exist"*

**Reply**: Changed
* * *
**L400** *The meaning of "this" is ambiguous. Can you not narrow it down with some single scattering comparisons?*

**Reply**: "this" has been replaced with "solar refraction" to clear up the ambiguity. For sure additional simulations could be performed that could narrow this down, but we are unable to isolate this effect with the simulations that we have already done. We definitely agree that this is interesting and as stated in the manuscript it is a subject for future study.
* * *
**L402 (and L471)** *higher → larger (see L470)*

**Reply**: Changed.
* * *
**L415** *Delete ", solar geometry, and atmospheric composition"*

**Reply**: We think it is important to be clear here. If we only state that this is average time is for a single wavelength the reader could incorrectly infer that this time is the time

to calculate all solar geometries and compositions at that wavelength, when that is not the case.
* * *
**L423** *The → the*

**Reply**: Fixed.
* * *
**L441** *Search the document for polaris\* and replace the 8 occurrences with polariz\**

**Reply**: Thank you, this has all been changed to be consistent.
* * *
**L448** *low → small*

**Reply**: Changed.
* * *
**L455** *Is refraction considered at all altitudes for SMART-G or does it "turn on" when the altitude is low enough (e.g. 11.5 km)?*

**Reply**: It is considered at all altitudes, as you can see in Figure 9 the refraction ratio is not 1 at the higher altitudes. The cause of this is still somewhat of a mystery.
* * *
**L477** *university → University*

**Reply**: Fixed.
* * *
**L720** *No need to provide second website and publisher in this reference and many others, or is this a new convention?*

**Reply**: We are also unsure about this but this is automatically generated from the copernicus AMT bibtex template so I assume if it is not correct it can be fixed during copy-editing.

---

## Author Comment (AC3)

**Responses to Referee 2 (Chris McLinden) on Behalf of the Authors**

We would like to thank the referee for their helpful comments and suggestions. Included below is each of the referee's comments (italics) followed by our reply.

**Specific Comments**
* * *
**Line 36** *Incorrect reference - this paper should be cited doi:10.1029/2009JD012488*

**Reply**: Thank you, this has been fixed.
* * *
**Table 1** *what was the rationale for choosing these combinations? Are these indicative of OSIRIS, SCIA, ALTIUS, etc...?*

**Reply**: These combinations were initially chosen as representative for ALTIUS, but they are fairly representative for near polar sun-synchronous orbits that have an equitorial crossing time not near dawn/dusk. We have added a statement to this effect in the manuscript.
* * *
**Table 1** *What about using SZAs through sunrise/sunset (e.g., 85-95) – some useful information can be gleaned analyzing limb observations through this period. See, e.g., Atmos. Chem. Phys., 8, 5529–5534, 2008*

**Reply**: We definitely agree that this would be a very useful exercise and something that should be done. Right now it is beyond the scope of what we set out to do and would have to be done in future work. Previously we had a statement in the conclusions that more extreme solar zenith angles should be checked, but the way it was written it could be interpreted that was specifically for checking solar refraction. We have modified the statement in the conclusions to better emphasize the importance of larger solar zenith angles.
* * *
**Line 369** *"Both SASKTRAN (HR and MC) and GSLS make the assumption that V is identically 0" ... I assume this is what is assumed here, and not a limitation of the models. That is, they can handle a 4x4 phase matrix. Please clarify.*

**Reply**: You are correct that this is not a limitation of either technique, there is nothing in the equations themselves that prevent you from using a 4x4 phase matrix. However, both models, at least right now, do not have an option that you can turn on/off to switch between a 3x3 or 4x4 phase matrix. We have added the statement "The approximation is

not fundamental to the method of solution used by SASKTRAN (HR and MC) and GSLS, but currently the models do not have an option to remove it."
* * *
**Table 2 / line 245** *what is the aerosol OD? Provide at a reference wavelength, or add to table 2. I assume the extinction of number density profile is provide with the other reference material? If not, please add.*

**Reply**: Yes the aerosol profile is provided as part of the reference data. The vertical optical depth at 675 nm has been added to the manuscript.
* * *
*In a future work it would be good to compare under more demanding conditions, such as larger SZA and higher aerosol loadings and/or clouds, non-Lambertian surfaces*

**Reply**: We agree on all these points, the conclusions have been modified to include all of these points as potential areas of future work.
* * *
*Is it useful to compare the multiple-scattered component by itself (I – Iss) ?*

**Reply**: This is something that we thought about for a while during the comparison process, with the motivation being that $I_{MS} = I - I_{SS}$ is the actual difficult quantity to calculate. The problem was that in various scenarios $I_{MS}$ is quite small, making % difference not a perfect metric of the observed differences. In the end since $I_{SS}$ agrees very well between all of the models and $I$ is the actual quantity of measurement interest it was decided to do the comparisons with $I$.
* * *
*Mention some general findings related to the 1700 nm comparisons where the signal would be dominated by aerosol scattering.*

**Reply**: This is an interesting point. When we initially did the comparisons we found that the differences at longer wavelengths were comparable to 675 nm, and thus the decision was made to not go farther out since it becomes increasingly difficult to execute some of the models to good precision at longer wavelengths. But based on this comment we went back and looked at why this would be the case and we found that even at 675 nm, for some of the solar geometries and high albedo, aerosol scattering can be  75% of the signal at the aerosol peak. We have added the statement "We have found no differences that are indicative of differences in stratospheric aerosol scattering. Differences at longer wavelengths (not shown) are comparable to differences at 675 nm, and at 675 nm aerosol scattering can make up 75% of the observed signal in the forward scatter high albedo case."

---

## Author Comment (AC4)

**Responses to Sergey Korkin on Behalf of the Authors**

We would like to thank Dr. Korkin for both being interested in the manuscript and taking the time to provide clear and helpful feedback. Included below is each of the referee's comments (italics) followed by our reply.
* * *
*Overall, it is a great paper with valuable reproducible benchmark results (to the best of my own knowledge, reported first time) written by leading experts in RT numerical simulations. Below, I dare to provide some criticism, questions, and comments, the number of which may seem high. But it only indicates that the paper is interesting and reports new, practically important, and reproducible results. I'll be happy if the authors will address some of my comments, but will not argue if they skip them all.*

**Reply**: We are happy that you found the manuscript interesting, and once again want to thank you for taking the time to provide helpful suggestions. We have tried to take into account as many of them as we can.

**General Comments**
* * *
**Section 2 "Model Descriptions"** *For each model, please indicate if it is publicly available and provide weblinks. E.g., MYSTIC libradtran.org is publicly available without limitations, while distribution of SCIATRAN may require registration, etc.*

**Reply**: Providing the website for each of the models is a good idea and so we have added that for the models that do have a website. In terms of licensing/availablity/etc. this can be fairly technical and subject to change so we prefer to leave that information out of the manuscript since any potential user can find it on the model website.
* * *
*For each model, it would be very helpful to provide a ready to use input (with short instructions), so that an interested user could download and quickly run a model and be able to independently reproduce the reported results;*

**Reply**: We agree that this would be quite useful. It is quite challenging in that it is a fair amount of work and that some of the models are not publicly available (at least without request) and may not have installation support. Many of the models do come with examples, or have examples available online or in documentation, and that would be a natural place to put something like this. We as the model authors can put providing something like this in the model documentation on our ever expanding list of improvements to make.
* * *
*Numerical results (benchmarks) are the most useful part of the paper. However, it is not clear from the beginning how to get the numbers. A simple python driver to read the*

*provided netcdf file would also be helpful due to the amount of the data. I would recommend providing structure (tree) for the netcdf file (e.g. in Appendix).*

**Reply**: In the introduction we have added a statement saying that the model results are publicly available and where to find them. Some more information on the data structure is a good idea, we don't feel that the proper place to put it is inside the paper however. Since a separate reference exists for the dataset on Zenodo we can add the netcdf file structure there.
* * *
*Different instruments are discussed in the Introduction; however, their measurment accuracy is not indicated. The authors are talking about 0.1, 0.5, 1% error in simulations (e.g. Abstract & Conclusion). However, it is not clear if this level of error is too good or just right or insufficient. Since ratios of intensities are often used, one should keep in minds that sufficient level in simulated intensity may or may not be sufficient for the ratio.*

**Reply**: This is a very good point. The 1% level that we present is not chosen based on any specific application or need, but rather if a model differs by more than 1% then we have high confidence in saying that something is different about this model. The required level of error is going to be heavily dependent on the specific application. With the ratio example it could be the case that any given RTM is 5% off in the radiance calculation, but when you take a ratio the errors completely cancel out. We have added a statement to the (ever expanding) future work part of the conclusions that how the observed differences affect different applications could be studied.
* * *
*Fig.2: definition of angles is unclear. Specifically, is the zero relative azimuth correspond to forward or backward scattering? Same for the zenith angle 'theta' (bidirectional arrows are confusing). It would be helpful to show observation of another tangent point (I believe the observer will change location, so that local normal at the tangent point would not move);*

**Reply**: We see how this can be ambiguous. In the figure caption we have added a statement saying that 0° solar scattering angle is perfect forward scatter, and 0° solar zenith angle is the sun directly overhead. We believe that adding in a second line of sight to the figure makes the figure too crowded since it is already quite busy.
* * *
*Across the text: does the multiple scattering include the first one? Or multiple literally means second and higher?*

**Reply**: Taking a closer look at this the answer is apparently both. We have changed the wording so that multiple scatter refers to second order and higher, and that when we refer

to the total radiance we say that multiple scattering is enabled or included.
* * *
*Line 230: 11 wavelengths. Why all 11 are needed? It would make since to test the codes for the shortest band, longest band, and something in between (like, Lambertian surface is defined for black, gray and white cases – only three). If an RT code works for 3 mentioned bands, why it would not work for the rest? Does any of the mentioned 11 bands have some interesting features (e.g., peculiarities in O3-profile), not presenting in other bands? Leaving only 3 necessary bands would make it easier to represent the results while pursuing "the purpose of brevity" (line 286).*

**Reply**: The initial idea was if there was an issue in an RTM due to aerosol scattering then we might be able look at a long spectral gradient to isolate the effect. The same idea applies for ozone absorption. In the end we found no deviations that were indicative of aerosol scattering or ozone absorption problems and so many of the wavelengths were repetitive and not used. However, we have the data, and maybe a future model will find these longer wavelengths useful. Based on the comments of one of referees a brief statement about the longer wavelengths was added to the manuscript.
* * *
*Section 2.2.5. is important but very confusing... First, in the parenting Section 2.2, two approaches are discussed: forward and backward. What about local estimation - is it similar to backward (excatly the same... completely different...)? Further, MYSTIC "used the backward MC method" (line 171). However, in lines 210-214, only Siro and SASKTRAN-MC are mentioned as based on backwards technique, while MYSTIC is mentioned one paragraph above (different technique?) Condition "not if scattering actually occurs" (line 211) is unclear. Neither is "a desired noise floor" (line 213) term – does that mean level of MC noise? Finally, description of the SMART-G feature (line 215) seems too brief to understand – please either elaborate or reformulate or maybe refer to the SMART-G section. Overall, differences between MC techniques deserves a separate paper, and it is great the authors decided to summaries them in one paragraph. But such a paragraph should be written with extreme care.*

**Reply**: We thank Dr. Korkin for recognizing our challenges in trying to write such a section. Your confusion is very valid, and we think at least some of it results from what is normally assumed in the limb radiative transfer community compared to the non-limb radiative transfer community. All of the MC models here are using a backwards method, the forwards method is actually not mentioned anywhere in the manuscript. Local estimate is a technique that would be used within a forwards method but it is not applicable to the backwards method, and so not relevant here. The difference that we are interested in is actually a subtle difference within the backwards method, i.e., the difference between assuming a scatter occurs and using MC to find the point of scattering, or traversing a photon through the atmosphere and using MC to determine if a scatter occurs at each interface. To try and make this more clear when we introduce MC we have explicitly stated that all models here are using a backwards technique, and also renamed the section from "Differences in the

Monte Carlo Methods" to "Differences in the Backwards Monte Carlo Methods".
* * *
**A whole paragraph around line 255, where polarization discussed** *what is the frame of reference that defines the Q & U components?*

**Reply**: We have added "We define a cartesian coordinate system $z$ (vertical), $x$ (South), and $y$ (East), the reference frame for the Stokes vector is then the plane spanned by the $z$ and line of sight directions. In this frame $Q$ is the "vertical" polarization and $U$ is the "horizontal" polarization."
* * *
*For aerosol scenario, it would be helpful to plot the phase matrix vs scattering angle and tabulate expansion moments for deterministic RT codes.*

**Reply**: The phase moments (and the phase matrix) are provided in the supplemental data, they would be quite long to tabulate within the manuscript itself. We are hesistant to add another figure to the paper since the referee comments indicated the paper is long as it as.
* * *
**Section 4.3** *how percentile deviation for DoLP is defined? At neutral points, DoLP $\to$ 0 and delta DoLP would be Inf. "LPO" - orientation wrt what reference plane?*

**Reply**: We only present absolute differences of DOLP for the exact reason that you describe, so defining percent difference is not required. The LPO is defined such that it is 0 if $U$ is 0, so this would be relative to horizontal polarization. In the polarization section we added the definition of the reference plane for $Q$ and $U$
* * *
**Table 4: Timing** *does the scaled runtime account for different number of CPU cores in addition to difference in CPU themselves? E.g., if one runs SASKTRAN on 6 cores (instead of 12 as indicated), will the result be 2 x 1909 = 4000?*

**Reply**: Yes it does, we have added a clarifier in the text which should make this clear.

**Minor Comments**
* * *
**Title** *I've never met the word "scalarial" and would suggest to use "vector" instead of "vectorial". The word "vector" is actually used in the text "vector test cases" (line 7), "vector modes" (line 11), and other places in the paper including list of references; See also*

*line 102 for "... vectorial radiance...".*

**Reply**: Oddly enough it appears that "vectorial" is the correct adjective form of "vector", and "scalar" is the correct adjective form of "scalar". You are correct that they have been used inconsistently throughout the manuscript, and we have changed some instances of "vector" to "vectorial", which might not have been the intended effect of this comment...
* * *
**In the Abstract** *"fully spherical" sounds confusing and should be either explained or reduced to "spherical", while the word "fully" should be introduced and explained later.*

**Reply**: We agree and this has been changed.
* * *
**Line 131** *"0.2% accuracy" – I believe, you are targeting 0.2% \*error\*...*

**Reply**: In this case we don't think there is a difference since there are no random errors involved.
* * *
**Fig. 1** *"SSA" could be confused with single scattering albedo. Maybe SCA (scattering angle) sounds better?*

**Reply**: We agree this can be confusing. Since Table 1 was actually the only place the abbreviation SSA was used we just replaced it with the full "Solar Scattering Angle".
* * *
**Lines 50 & 92** *"approximative" – please confirm the word is correct. In line 103, I see "approximate" twice which sounds better.*

**Reply**: Both of these instances have been changed to "approximate".
* * *
**Line 78 and across the text** *"polariSed" is correct, but "polariZed" is used more oftet, including this paper: 6 times for polariSed, 15 for polariZed (as Ctrl+F in Acrobat shows me)*

**Reply**: Thank you, everything has been changed to be consistent.
* * *
**Line 98** *"...single plane parallel solution" I was not able to understand what it means.*

**Reply**: The previous paragraph refers to the approximate spherical technique, where multiple plane parallel solutions are used. This refers to the case where only a single plane parallel

solution is used, likely at the tangent point. We have changed a little bit of wording around here so hopefully this comes across.
* * *
**Lines 104-106** *"The number of solar ... (2015)" – I would suggest to reformulate as "The number of solar zenith angles FOR the multiple scatter source function depends on ..."*

**Reply**: We see the point that the current wording is a little terse but the proposed change can give off the impression that the calculation is coupled in some way, or that this is an angular discretization within a single calculation. The actual method involves separate calculations of the multiple scatter source at each of the solar zenith angles that are independent.
* * *
**Line 143** *"The comparison shown... SCIATRAN" – would it be better to reformulate it like this: "SCIATRAN uses fully spherical mode in the shown comparison". In general, I have a feeling that the authors use passive voice too often.*

**Reply**: Thank you this has been changed.
* * *
**Line 160** *what is "three FULL dimensions"?*

**Reply**: Point taken, full is not needed here and has been removed.
* * *
**Line 191** *"...wavy interface () or any surface spectral BRDF boundary condition" – I would suggest simplification: "...ocean and land." It is clear that ocean has waves while land reflection is in general non-Lambertian and spectrally dependent.*

**Reply**: While it is clear that oceans have waves and that land is generally non-Lambertian it is important to mention what features the model can actually handle. Not all models included here can handle non-Lambertian BRDFs for example.
* * *
**Line 204** *"While all four models listed above use Monte Carlo" – the sentence is excessive: it is clear from the Section title one line above.*

**Reply**: Sentence has been changed to "There is a subtle difference in the way the backwards Monte Carlo method is implemented that can be noticed in the subsequent comparisons. "
* * *
**Lines 254 & 310** *please confirm "differing" is the right word*

**Reply**: On L310 "differing" was odd so it was changed to "different", on L254 there were

two instances of "differing" and we changed one to "different".
* * *
**Line 261** *what is "atmospheric state"? Please list all parameters.*

**Reply**: We have added "(number densities of various species, temperature, and pressure)"
* * *
**Line 271** *what it means to compare models "more directly"? The authors are likely talking about making input as close as possible.*

**Reply**: This is correct, we have reworded this to "To better harmonize the treatment of the input data"
* * *
**Line 372** *"It is possible...."- this should be either further explained or, better, dropped.*

**Reply**: The point of this is really just to motivate future work. We have added a little bit of more explanation saying that this relates to the coupling of circular and linear polarization, as well as directly stated here that it is something that could be studied in the future.
* * *
**Line 435** *last sentence in the section is very confusing. It sounds like SMART-G simulates only single scattering in spherical geometry. Please reformulate (or drop, because all necessary reference to SMART-G are in References).*

**Reply**: Statement has been reworded at "SMART-G improves the relative precision of the calculation by forcing scatter events to happen on the first order of scatter (similar to SASKTRAN-MC and Siro which force scatters on all orders), and it appears that this is sufficient to obtain reasonable precision in all scenarios."
* * *
**Line 444-445** *it is strange to "agree to a level better than the precision of calculation" (because difference in results cannot be treated accurately at that level). Would it be better to say something like "the codes agree within the expected accuracy?"*

**Reply**: We agree that this statement was odd. We changed to say that they agree at the level of the precision of the calculation instead of better.
* * *
**Line 455** *the meaning (not the cause) of "discontinuity in radiance" in SMART-G remains*

*unclear to me, but maybe I am just a bit tired by the end of the paper...*

**Reply**: We changed discontinuity to "jump".
* * *
**Line 466** *"...upcoming ALTIUS mission which is linearly polarized" – sounds like the mission is polarized. I would cut the sentence at "…. ALTIUS mission."*

**Reply**: Thank you, that is better.